# Enabling Space-Aware Service Discovery Model in Home Networks through a Compatible Extension to mDNS/DNS-SD

Chun-Feng Liao [1,2,*] and Yu-Jou Weng [1]

1  Department of Computer Science, National Chengchi University, Taipei 116, Taiwan; 108703042@nccu.edu.tw
2  Program in Digital Content and Technologies, National Chengchi University, Taipei 116, Taiwan
*  Correspondence: cfliao@nccu.edu.tw; Tel.: +886-2-2939-3091-62297

**Abstract:** The advent of the smart home, fueled by the rapid emergence of tiny embedded IoT devices and sensors, brings the consideration of space and location information to the forefront of service discovery. In a smart home environment, spaces often have composite capabilities and complex structures. Since residents may have varying preferences based on their location within different spaces, traditional service discovery results may be suboptimal without considering the spatial context. This paper introduces a seamless approach to integrate space and location information with mDNS/DNS-SD, a widely utilized service discovery protocol in home networks. We provide a formal specification for this approach, develop a prototype, and perform a series of experiments to evaluate the efficacy and potential of the proposed method.

**Keywords:** service discovery; home network; mDNS/DNS-SD; ambient; space

## 1. Introduction

In recent years, the rapid development of information technology in hardware and software along with the popularity of the Internet has led to the emergence of many new internet of things (IoT) applications. The trend of IP convergence facilitates innovative services to be created based on interconnected IoT devices. In this context, the smart home is one of the most popular IoT services; it consists of connected IoT devices scattered within a wired or wireless LAN (local area network), namely, a home network, where the devices collaborate to sense environmental changes and user behaviors, which can be used to anticipate user intentions and then to satisfy their needs. In a home network, it is essential to employ a "service discovery" mechanism to coordinate the detection, composition, and interaction of IoT devices.

Among various service discovery protocols, mDNS/DNS-SD (multicast DNS/DNS-service discovery) is standard-based, compact, and widely deployed in smart environments [1]. mDNS/DNS-SD is a combination of multicast DNS [2] and DNS-service discovery protocol [3]. Unlike traditional DNS, mDNS provides a domain name to the IP resolution service without a centralized DNS server. It provides DNS service via IP multicast, with the multicast address of 224.0.0.251 (IPv4) or [FF02::FB]:5353 (IPv6) in a LAN. On top of mDNS, DNS-SD specifies how the DNS resource records are structured to provide service discovery functionalities. This decentralized feature makes it more suitable for smart environments such as smart homes, where it is infeasible to deploy a dedicated server. mDNS/DNS-SD is currently being considered as the official discovery mechanism in the W3C web of things (WoT) discovery specification [4]. Hence, mDNS/DNS-SD has the potential to become a mainstream service discovery mechanism for smart environments, and it is the focus of this paper.

In a smart environment, the residents tend to have different preferences when they are located in different spaces. Consider the smart home environment shown in Figure 1: when a user is in the study room and the user needs a display service, it makes no sense if the

system provides the display service using a device located at the kitchen. Spatial (location) information has been identified as an essential context type in smart environments in the literature [5]. However, like most existing service discovery mechanisms, mDNS/DNS-SD are not aware of spatial information. mDNS/DNS-SD can discover connected IoT devices by name and by service type. It is also possible to perform sophisticated selection by attaching filters as key = value pairs in the DNS resource records [6]. A straightforward approach would involve associating spatial (location) data with a textual location label (e.g., "kitchen"). Service requests can then be matched based on these location labels corresponding to device labels, with these labels being embedded in the DNS resource records.

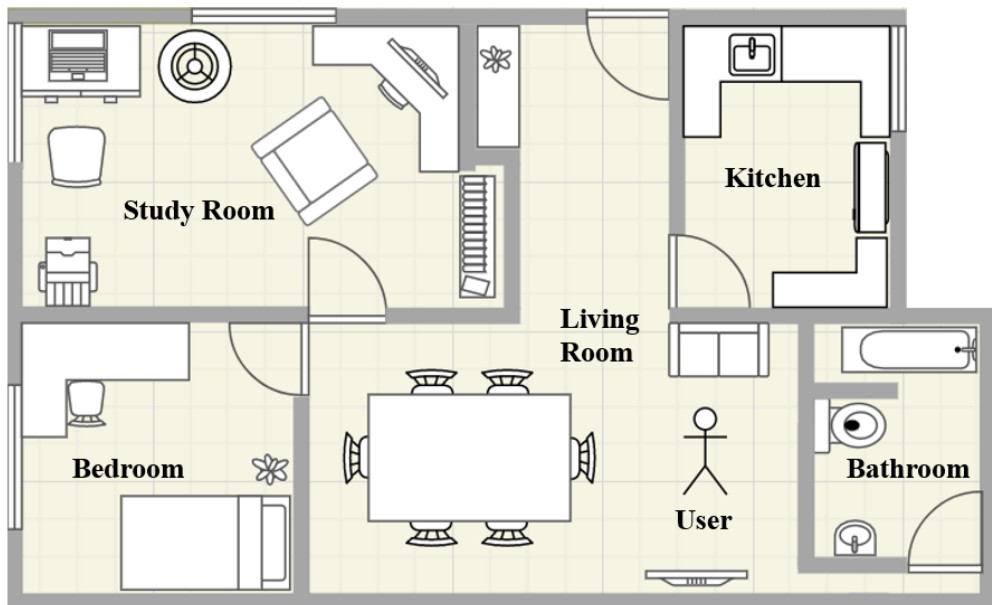

**Figure 1.** The floor layout of a smart environment.

The key = value pair format, unfortunately, lacks the expressiveness to effectively represent spatial partitioning and containment relationships. In essence, spaces can be composite, meaning one space can be nested within another, and they can possess intricate structures. Figure 2 illustrates how the smart home environment, shown in Figure 1, is divided into several subspaces. In this depiction, a single space can simultaneously encompass another space and be contained by a third. For instance, in Figure 2, the living room, labeled as 'D', encloses spaces 'H' and 'I'. Concurrently, the living room is enveloped by space 'C'. As such, the smart dining table's location in the living room is identified as 'H'. Yet, it would also be accurate to state the dining table is situated in 'D', 'C', or even 'home', considering 'H' is a subset of 'D', 'D' is a part of 'C', and 'C' falls within 'home'. Clearly, denoting these layered relationships using mere key = value pairs is not straightforward.

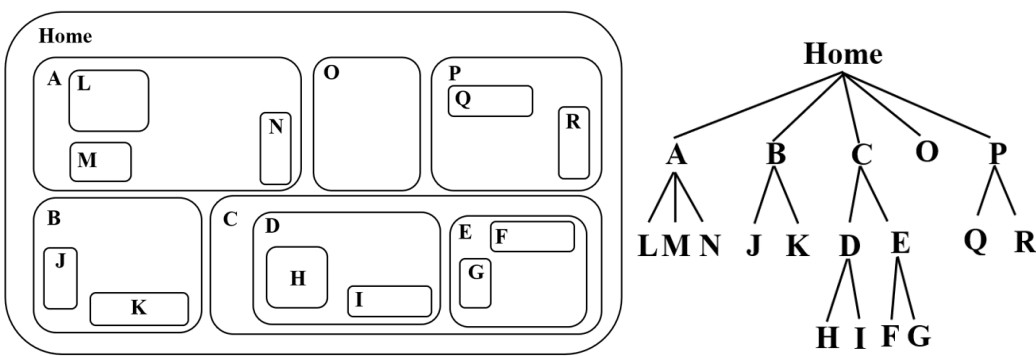

**Figure 2.** (**Left**) One possible space partitioning of the smart environment shown in Figure 1; (**Right**) the ambient structure tree of this space partitioning.

Based on the preceding discussions, the central aim of this research is to develop a space-aware extension for mDNS/DNS-SD. The contributions of this paper are twofold: First, we formally introduce the syntax and semantics of spatial abstractions, incorporating both a dedicated data structure for representing spatial relationships and an expression for specifying device locations and search criteria in service discovery. Second, we rigorously define the behaviors of mDNS/DNS-SD and propose a non-intrusive method to integrate spatial expression, thereby enhancing its spatial awareness capabilities.

The remainder of this paper is structured as follows: Section 2 reviews the state of the art and related work. Section 3 formally details the spatial abstraction concepts employed in this study. Section 4 delineates the design of the proposed extension, elucidating how it non-intrusively augments mDNS/DNS-SD with spatial awareness. Section 5 evaluates the design through a series of experiments, discussing both the limitations and lessons learned. Finally, Section 6 offers conclusions and outlines avenues for future research.

## 2. Related Works

In recent years, the rapid development of internet of things (IoT) technology has seen mDNS/DNS-SD emerge as IETF's service discovery solution for IoT. Siljanovski et al. demonstrated the feasibility of successfully implemented mDNS/DNS-SD on embedded systems (ATMega1284p, 8 MHz, 128 KB flash, and 16 KB RAM) [7]. Despite its high compatibility and widespread use, mDNS/DNS-SD suffers from a lack of explicit guidelines concerning endpoint communication behavior, leading to considerable performance variances across different implementations [8]. As a result, much of the existing research aims to enhance efficiency and reduce network traffic. For example, Mahmoud et al. proposed configuration mechanisms designed to minimize both packet counts and memory usage [9]. Lee et al. introduced DNSNA (DNS name autoconfiguration) as an innovative alternative to mDNS/DNS-SD, optimizing efficiency by leveraging IPv6 neighbor discovery and the dynamic host configuration protocol and ensuring security through near field communication [10]. DNSNA has been shown to reduce packet numbers by 60.8% and overall packet volume by 97%.

Another notable challenge in utilizing mDNS/DNS-SD within IoT environments is its requirement for IP-based connectivity across all devices. To circumvent this limitation, our research team has developed mechanisms to facilitate seamless device management in both 6LoWPAN (IPv6 over Low Power Wireless Personal Area Networks) [11] and BLE (Bluetooth Low Energy) [12] using mDNS/DNS-SD. These advancements render our proposed approach compatible with two of the most widely adopted low-power and lossy network (LLN) standards in the IoT landscape. In application-specific domains, research has explored the use of mDNS/DNS-SD in intelligent transportation systems (ITS) [8], smart cities [13], and microservices management [14].

Service discovery serves as the method by which a device or user locates a target device for interaction, a functionality that has become increasingly important in internet of things (IoT) settings [1,15]. Typically, this process commences with the issuance of a query containing a set of criteria that the target device must satisfy. The quality of the discovery results is intrinsically linked to the specificity of these search criteria. For instance, service type (e.g., printer, display, computer, and temperature sensor) is a commonly used criterion for service discovery. When multiple devices meet the specified criteria, the resulting list can be prioritized based on user preferences [16,17]. Additional descriptive information about the services becomes crucial for effectively ranking these results. However, from the perspective of a service discovery protocol, mDNS/DNS-SD has been criticized for its limited capability to provide descriptive information about services. Addressing this limitation, Stolikj et al. identified two possible extension mechanisms: the first involves appending search attributes to the domain name, and the second employs the Key = Value syntax within TXT resource records to describe quality of service (QoS) attributes [6].

Accurate spatial awareness in smart environments such as smart homes is often contingent on the efficacy of indoor positioning techniques. While the majority of existing

indoor positioning systems prioritize identifying user locations, device localization has been comparatively overlooked. This oversight stems from the inherent challenges of accurately pinpointing device locations. Most current systems necessitate the attachment of either WiFi or a BLE beacon profile to each device [18]. However, these methods not only elevate operational costs but also consume excessive energy due to continuous monitoring, especially when devices are dispersed throughout various locations. Battery-powered devices introduce the logistical issue of frequent battery replacements, while devices connected to power lines are less likely to be moved, reducing the utility of location tracking. As a consequence, many systems rely on manual configuration and the updating of device locations [19], a limitation shared by our work.

The concept of an "ambient" serves as a valuable formal abstraction for modeling the space and the mobile services in the spaces [20]. Defined as a bounded space with computational capabilities, an ambient is inherently composite-capable. This means that ambients can be nested within each other, leading to complex spatial structures. Recently, the formal semantics of ambients have been extended to physical objects [21] and IoT environments [22].

To conclude, it is crucial to elucidate the distinct contributions of our research in comparison with existing studies. While current research trends largely aim to optimize efficiency and reduce network traffic in mDNS/DNS-SD implementations [9,10], extend these protocols for LLN [11,12], or focus on domain-specific applications, our work takes a divergent path. Specifically, our research focuses on enhancing the spatial search functionalities within mDNS/DNS-SD. To this end, we propose an innovative extension that seeks to elevate the quality of service discovery in home networks through the integration of spatial data. Furthermore, as mentioned above, Stolikj et al. [6] have proposed two separate strategies for augmenting QoS attributes. We employ both of these strategies to craft a more nuanced and effective service description. To elaborate, we augment the domain name with service types denoted as sub-types, aligning with their first recommendation. Concurrently, we incorporate spatial information within the TXT resource records, utilizing a key = value schema in accordance with their second recommendation. This dual-strategy approach facilitates a more comprehensive service discovery mechanism. Lastly, when it comes to the formal abstractions of ambient environments, our research distinguishes itself from prior studies [21,22] that primarily focus on modeling tangible objects or well-defined spaces. In contrast, our work places a premium on capturing the spatial relationships between different locations, integrating this aspect seamlessly into the existing service discovery protocols. This allows for a more holistic and applicable approach to service discovery, tailored to the complex spatial dynamics of modern home networks.

## 3. Spatial Abstractions

This section delves into the conceptual and notational foundations that underpin the discussions in the subsequent sections. First, we define the essential concept of a location, referred to as an "ambient", and explore the containment relationships among various locations. Following that, we formally define the expressions used to represent these locations and clarify the semantics of spatial matching.

### 3.1. Ambient

As mentioned, representing spatial relationships is not trivial. In this paper, we use "ambient" as the spatial abstraction based on the concepts of ambient containment and the ambient structure tree presented in our previous work [23]. Concretely, the "ambient" is a physical living space or one of its subdivided spaces. Consider the spaces in Figure 2: the spaces A, B, . . . , R are ambients according to such a definition. Note that spaces can be embedded within one another. This kind of containment relation is called ambient containment. The ambient containment relation between two ambients is denoted using $\sqsupseteq$. For instance, in Figure 2, $C \sqsupseteq D$ and $D \sqsupseteq H$ ambient containment is transitive, so if $C \sqsupseteq D$ and $D \sqsupseteq H$, then $C \sqsupseteq H$.

Let us denote the ambients of a smart environment $\varsigma$ as a set $A = \{a_1, a_2, ...a_n\}$ and denote the ambient containment relations ($\sqsupset$) between arbitrary $a_i$ and $a_j$ as a relation $C$, where $C \subseteq A \times A$. In this way, $(A, C)$ forms a rooted tree. Given that $a^{parent}$ is the parent node of $A^{child} = \{a_1^{child}, a_2^{child}, ..., a_n^{child}\}$ in the tree, then:

$$\forall a_i^{child} \in A^{child}, a^{parent} \sqsupset a_i^{child}. \tag{1}$$

As a result, we can define a rooted tree $Y^\varsigma = (A, C)$ that conforms to (1) to be the ambient structure tree (AST) of the space $\varsigma$. As an illustrative example, the right-hand side of Figure 2 is the AST of the space partitioning depicted in the left-hand side of Figure 2. We can further define some useful functions to facilitate the descriptions of our design in the following sections. First, $parentOf(a)$ can be used to acquire the closest parent node of $a$ in the AST. In addition, using $childOf(a)$, we can obtain the set of immediate child ambients of $a$. Finally, $childOf^*(a)$ and $parentOf^*(a)$ are used to obtain the set of all parent or child ambients of $a$ (exclusively). For example, in Figure 2, $childOf(Home) = \{A, B, C, O, P\}$; $childOf^*(Home) = \{A, B, ...R\}$. According to the basic property of the tree data structure,

$$a^x \in childOf^*(a^y) \Leftrightarrow a^y \in parentOf^*(a^x). \tag{2}$$

Consider the ambient C and G in the right-hand side of Figure 2; it is obvious that $G \in childOf^*(C)$ if and only if $C \in parentOf^*(G)$.

The abstract syntax tree (AST) presented in prior research [23] is employed for message propagation. In the scope of this paper, we integrate AST with the mDNS/DNS-SD service discovery protocol to facilitate space-aware service discovery model. Generally, when spatial information is applied to service discovery, it refines the search scope, which can enhance both the result quality and the discovery efficiency, for instance, if a user wants to find the TV in the living room (the ambient D in Figure 2). based on the previous definition, $t \in childOf^*(D)$, where $t$ symbolizes the TV's ambient. Consequently, there is no reason to explore the ambient $\alpha$ if $\alpha \notin childOf^*(D)$. In other words, rather than navigating the entire AST, only ambients D, H, and I warrant examination.

### 3.2. Spatial Expressions and Spatial Matching Semantic

We introduce a slash-delimited format, referred to as the spatial expression (SE), crafted to denote an ambient's location within the AST. Assuming that IoT devices are labeled with an SE corresponding to their location, the discovery process is initiated by a client issuing a spatial query expression (SQE) to all IoT devices. Should the SE match the SQE, the matching device is deemed to fall within the desired spatial scope of the discovery, prompting the device to send its SE as a spatial response expression (SRE) back to the client. For example, the location of the TV that located within ambient I (see Figure 1) can be expressed as a SE: $Home/C/D/I$. To improve the expressiveness of the SQE, two distinct wildcard symbols are introduced: $*$ and #. The $*$ serves as a single-level wildcard, capable of matching any individual ambient within the AST, while # functions as a multi-level wildcard, designed to match multiple ambients." For instance, both $Home/C/D/\#$ and $Home/C/*/I$ match $Home/C/D/I$. The main difference between SQE and SRE lies in their distinct uses, leading to different syntactical constraints: SRE is used to represent a specific location of an item, so it cannot have a wildcard. Conversely, SQE can use a wildcard to broaden the search scope

Given the context described above, we can now precisely define the semantics of a "spatial match". From the perspective of service discovery, the SQE is used to delineate the spatial scope of discovery. If there exist a location (ambient) $\theta$ within the set of ambients $\Theta_{SQE}$ specified by the SQE, and this location "contains" the ambient $\vartheta_{SE}$ specified by the device's SE, then a match occurs because the device bearing $\vartheta_{SE}$ lies within the spatial scope defined by $\Theta_{SQE}$. Formally, $\vartheta_{SE}$ matches $\Theta_{SQE}$ if

$$\exists \theta \in \Theta_{SQE} : \theta \sqsupset \vartheta_{SE}. \tag{3}$$

## 4. Design

As mentioned, this paper primarily aims to augment the capabilities of mDNS/DNS-SD service discovery by enabling searches based on spatial information. In this section, we first delve into the challenges of integrating spatial data into mDNS/DNS-SD service discovery. Subsequently, we present our proposed methodology, which leverages AST to encapsulate and convey this spatial information within mDNS/DNS-SD.

### 4.1. mDNS/DNS-SD Message Structure

Before delving into the intricate details of our design, it is crucial to closely examine the basics of mDNS/DNS-SD. As the name suggests, it is bifurcated into two primary sub-protocols, mDNS and DNS-SD, with DNS-SD built upon mDNS. Like various decentralized service discovery protocols within a local area network (LAN), mDNS utilizes IP-multicast—as the prefix 'm' in mDNS indicates. Utilizing mDNS, the discovering client multicasts a query message to all devices within the same LAN. Devices that receive this message and find a match with the query then send a response message back to the inquiring client.

The mDNS format draws heavily from the structure of a DNS message [24]. As depicted in Figure 3, DNS messages are bifurcated into two types: query messages and response messages. Beyond their headers, each message comprises one or more sections. The query message encapsulates both the question and additional sections, whereas the response message incorporates the answer, authority, and additional sections. It is worth noting that both the query and response messages include a questions section, but the query message does not contain answer or authority sections. Within the various sections, a data structure termed the resource record (RR) exists. This RR is further segmented into several distinct sub-sections such as name, type, info, and data. Service discovery processes are articulated using a range of RR.

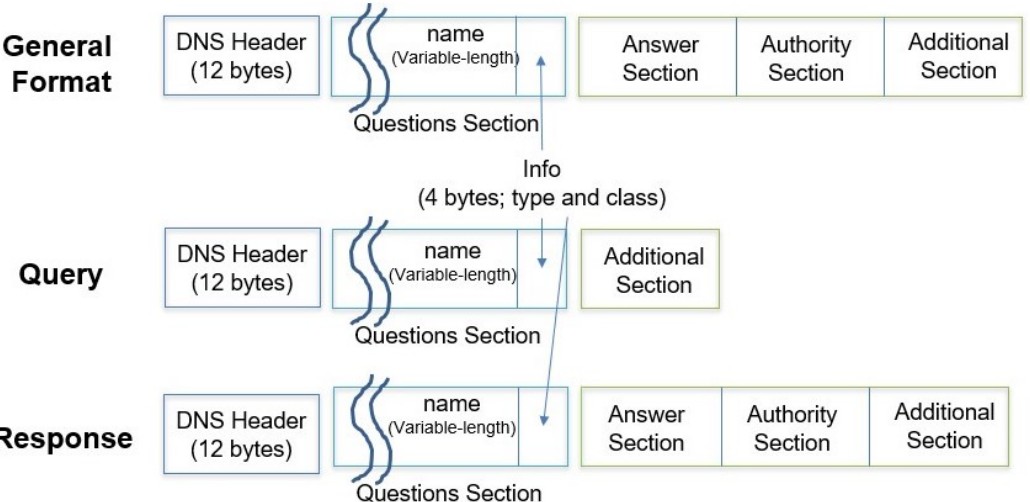

**Figure 3.** DNS message format.

There are five primary RRs: PTR, SRV, TXT, A, and AAAA. Specifically, the PTR RR identifies whether a given service type exists within the current domain; the SRV RR allows clients to inquire about the target service's port number and hostname; the TXT RR offers a space for embedding user-specified additional information; and the A and AAAA RRs are pivotal for retrieving the respective IPv4 and IPv6 addresses of the host.

DNS-SD facilitates service discovery based on mDNS. At its core, DNS-SD establishes a naming convention for local domain names, which is placed in the "name" sub-section of an RR, adhering to the format:

`<Instance>._<Sub Type>._sub._<Service Type>._<Transportation Layer>.<Domain>,`

where the term "service type" refers to the category of "internet services," which can be implemented using various application layer protocols, such as HTTP or FTP. On the other hand, "sub type" describes the specific functional aspect of the service. For instance, an HTTP-accessible binary light device operating over HTTP and TCP might be expressed as:

```
binaryLight1._binarylight._sub._http._tcp.local,
```

in which *binaryLight*1 designates the instance name, *_binarylight* symbolizes the sub type, *_http* denotes the Service Type, and *.local* identifies the domain.

In practice, relying solely on the service type and sub type can be insufficient for efficiently filtering the services that truly match a client's needs. Therefore, it is typical to employ key = value pairs to annotate specific services with more detailed attributes. These attributes aid service discovery clients in determining whether a particular service satisfies their requirements. For example, an additional key = value pair such as mode = binary can indicate light sources that only support an on/off function, whereas mode = dimming signifies light sources with adjustable brightness. This use of key = value pair metadata for more nuanced identification is generally termed as a QoS (quality of services) description. In DNS-SD, QoS details can be embedded in the TXT RR and included in the query's additional section, to be sent concurrently during the service discovery process.

### 4.2. Formal Expressions for Specifying mDNS/DNS-SD Behaviors

Most past research papers that discuss the behaviors of mDNS/DNS-SD have done so by directly presenting message content [25], utilizing flowcharts [26], or employing sequence diagrams [1]. While each method has its merits, there are inherent challenges: flowcharts can make it difficult to clearly present comparative improvements; the direct presentation of message contents might be overly detailed, thereby obscuring the overall design; and the use of sequence diagrams often necessitates additional text to clarify the design, making the explanation more cumbersome.

Since the interaction style among mDNS/DNS-SD endpoints is fundamentally similar to HTTP, with both adhering to the query–response pattern, we can design a query–response-style formal expression for DNS-SD's interactions, drawing on the notations developed in previous works Christian [27], Liao and Chen [28]. The query–response expression approach offers a concise presentation and enables higher-level comparisons and analyses of service discovery mechanism designs. Furthermore, the query–response expression can serve as a formal specification, allowing developers to implement the design in code with clarity and ease. In a subsequent section, we will provide detailed insights on how to translate the query–response expression to code at the implementation level. For now, we offer a brief outline of the formal expression syntax introduced in this study.

From a high-level perspective, the entire mDNS/DNS-SD system can be regarded as a tuple: $(P, R)$, where all mechanisms within this system are accomplished through a network of nodes $P$ transmitting DNS RR (denoted by $R$). Here, $c, \varsigma \in P$ represent the client (the querying endpoint) and the service (the responding endpoint), respectively. Each RR is characterized by a type $\tau$ (such as A/AAAA, PTR, SRV, or TXT), a name $\eta$ (formatted similar to a domain name), and a set of data $\delta$; thus, $R = \tau(\eta, \delta)$.

Next, we express the query–response relationship using a binary operator $\bowtie$, where the left side of $\bowtie$ represents the query and the right side represents the corresponding responses. Following the convention of Liao and Chen [28], we use the symbol $|$ to combine alternative responses, $\varnothing$ to represent an empty set, and $\perp$ to signify no response.

From these definitions, interactions between mDNS/DNS-SD nodes can be described by the following expression:

$$c.\Sigma_m R^{query(m)} + \Sigma_n R^{additional(n)} \bowtie \varsigma.\Sigma_i R^{answer(i)} + \Sigma_j R^{authority(j)} + \Sigma_k R^{additional(k)} | \perp, \quad (4)$$

where $m \geqq 1$ and $n, i, j, k \geqq 0$. The left side of Equation (4) represents the DNS query message sent by client $c$, containing $m$ query RRs and $n$ additional RRs to supplement query information. As depicted in Figure 3, the query message consists of the questions

section and the additional section. The questions section may contain one or more RRs, all without data, i.e., $R^{query(m)} = \tau_m(\eta_m, \varnothing)$, and the additional section may contain zero or more RRs; thus, $R^{additional(n)} = \tau_n(\eta_n, \delta_n)$.

Similarly, the right side of Equation (4) signifies that the responding endpoint $\varsigma$ will respond with zero or more answer, authority, or additional RRs, where $R^{answer(i)} = \tau_i(\eta_i, \delta_i)$, $R^{answer(j)} = \tau_j(\eta_j, \delta_j)$ and $R^{additional(k)} = \tau_k(\eta_k, \delta_k)$

### 4.3. Efficient Space-Aware Service Discovery Model with mDNS/DNS-SD

The traditional service discovery process implemented by mDNS/DNS-SD unfolds in four distinct steps. In the first step, a client $c$ initiates a service discovery request:

$$c.PTR(\eta_{stype}) \bowtie \varsigma.PTR(\eta_{stype}, \eta_{instance}). \tag{5}$$

Here, the request primarily consists of a PTR message, with $\eta_{stype}$ encompassing the service type, service sub-type (such as the application layer protocol, HTTP), and transport layer protocol (e.g., TCP). The right-hand side of $\bowtie$ outlines the response from a matched service $\varsigma$, which also utilizes a PTR message, encapsulating both $\eta_{stype}$, corresponding to the query message, and this service's specific instance name, $\eta_{instance}$.

In the subsequent step, we seek to identify the domain name, port, and TTL (time to live) of the matched services using its instance name. Formally, this is expressed as:

$$c.SRV(\eta_{instance}) \bowtie \varsigma.SRV(\eta_{instance}, (\eta_{host}, port, ttl)), \tag{6}$$

where $\eta_{instance}$ denotes the instance name of the matched service, while $\eta_{host}$, port, and ttl represent the domain name, TCP port, and TTL of that service, respectively.

Following this, the client can retrieve additional information from $\varsigma$ through a TXT message, utilizing $\eta_{instance}$:

$$c.TXT(\eta_{instance}) \bowtie \varsigma.TXT(\eta_{instance}, \delta_{info}). \tag{7}$$

As per DNS-SD specifications, the data accompanying the TXT message ($\delta_{info}$) must be rendered in a key = value format and specifically encoded. This key = value structure's adaptability provides opportunities for integration with other protocols, including the spatial expressions, SQE and SRE. In this manner, Equation (7) can be restructured as follows:

$$c.TXT(\eta_{instance}, \vartheta_{SRE}) \bowtie \varsigma.TXT(\eta_{instance}, \Theta_{SQE} \cup \delta_{info}). \tag{8}$$

This modification signifies that the SQE $\Theta_{SQE}$ is embedded into the TXT RR, and the response message from the matched service will also incorporate an SRE.

Finally, to actually access the service, the client must discern the corresponding IP for $\eta_{instance}$. This is achieved through either an A (IPv4) or AAAA (IPv6) query:

$$c.A(\eta_{host}) \bowtie \varsigma.A(\eta_{host}, ip). \tag{9}$$

From Equations (5) to (9), it can be observed that the original mDNS/DNS-SD requires at least $4(n + 1)$ messages to complete a search on a network with $n$ nodes. Moreover, while mDNS provides the "questions requesting unicast responses" mode, there is no compulsory usage of this in DNS-SD. As a result, if the unicast response is not used, the number of messages jumps to $4(n^2)$, representing a direct increase in order of magnitude. This is highly inefficient and can easily lead to the paralysis of the local network. In light of this, we adopt the optimization approach proposed by Mahyoub et al. [26] and modify the space-aware service discovery model process as follows:

$$c.PTR(\eta_{subtype}) \bowtie \varsigma.PTR(\eta_{subtype}, \eta_{instance}) \tag{10}$$

$$c.SRV(\eta_{instance})+TXT(\eta_{instance}, \vartheta_{SRE}) \bowtie \tag{11}$$
$$\varsigma.SRV(\eta_{instance}, (\eta_{host}, port)) + TXT(\eta_{instance}, \Theta_{SQE} \cup \delta_{info}) + A(\eta_{host}, ip)$$

This method differs from Equations (5) to (9) in that it omits the TXT and A (or AAAA) queries and instead directly places TXT and A (AAAA) into the additional section of the SRV response message. In this way, and in conjunction with unicast response, a single query requires only $2(n+1)$ messages at a minimum.

There are two potential issues with this method. First, as can be inferred from Equations (10) and (11), the SRV response packet will be larger than the original (as in Equation (6)). Despite this increase in size, experimentation has shown that this method can still save significant traffic compared to the existing mechanism. Second, this optimization assumes that $\eta_{instance} = \eta_{host}$; otherwise, it cannot function properly. However, this assumption is likely reasonable in mDNS, as, according to the spirit of mDNS, nodes on the network have the right to propose their own domain names.

## 5. Evaluation

In this section, we present the implementation of our proposed space-aware service discovery model mechanism, specifically focusing on its application within a WoT (web of things) context [29]. Additionally, we detail the experimental results, highlighting the impacts of this approach on the overall efficiency and effectiveness of the service discovery process.

### 5.1. Implementation

First, we demonstrate the feasibility of our proposed design by creating a space-aware service discovery model module. Though designed for general use, it can be integrated into various situations, including utilization by WoT devices. This prototype has been developed atop Node.js, which can run on typical embedded computer such as Raspberry Pi. To replicate a home network environment, a WiFi router is employed to establish a local area network (LAN). The router's configuration is modified to support multicast, enabling the WoT client to discover WoT devices via mDNS/DNS-SD. To realize the space-aware discovery mechanism described previously, each WoT device communicates its IP address and spatial expression through mDNS/DNS-SD. Within this system, every device has an attribute named "status" to signify its current condition. For instance, a service sub type "light" may have states like "on" or "off". Additionally, each WoT device is equipped with an attribute called "location", representing its physical location through a spatial expression, and a method named "toggle", allowing for state alteration.

Figure 4 illustrates the overall architecture of the prototype system, comprising a collection of WoT devices pre-configured with location information using spatial expressions. These devices offer various services, such as reading sensor values or toggling light switches, through HTTP-based operations. The device controller module encapsulates the overall logic of WoT. It orchestrates service discovery through the space-aware service discovery model module, manipulates sensors and actuators via internal GPIO, and interacts with WoT clients through the HTTP server. Initially, the functions and locations of the WoT devices remain unknown to the client. To uncover the devices that meet its requirements, the WoT client initiates a space-aware service discovery model process. This procedure is managed by the space-aware service discovery model module, which operates identically in both clients and devices. Upon completion, the WoT client not only identifies the appropriate devices but also ascertains how to access them, including the URL, IP address, and port. The client can then interact with the selected device using HTTP. The service discovery module is divided into two components. The first, referred to as the capability description module, ensures the device's functionality and is essential for all standard service discovery processes, though it is not the primary focus of this paper. The second

component, named the spatial expression handler module, is responsible for interpreting and matching spatial expressions.

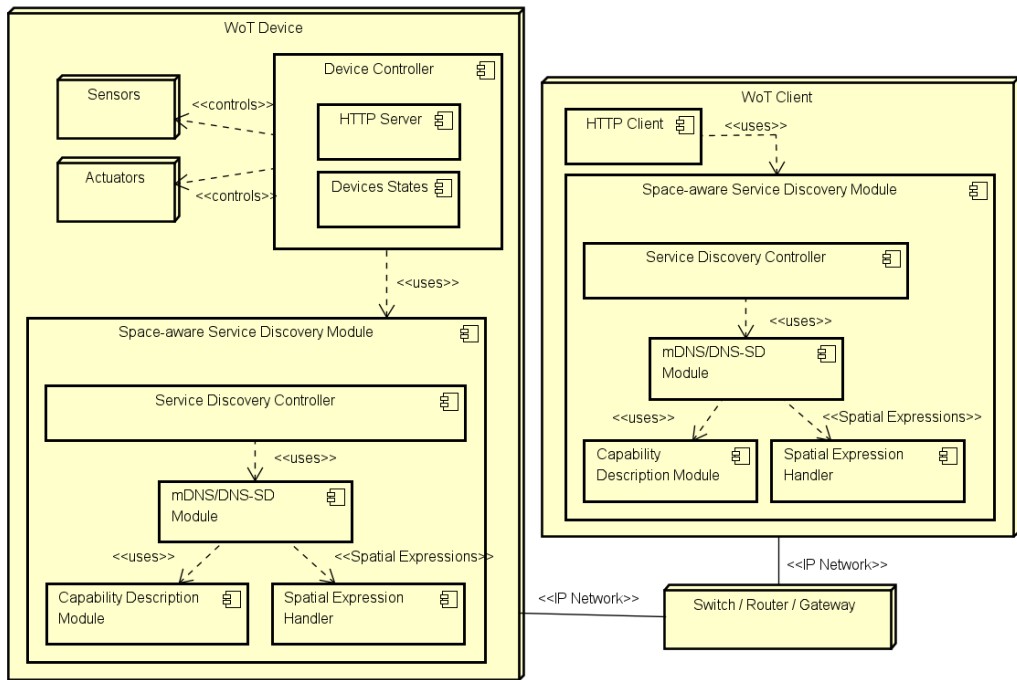

**Figure 4.** Overall architecture of the prototype system.

In the course of our implementation, we leveraged multiple open-source libraries. The space-aware service discovery model module, a shared component between the WoT client and WoT device, was developed using multicast-dns (https://github.com/mafintosh/multicast-dns, accessed on 26 August 2023), qlobber (https://github.com/davedoesdev/qlobber, accessed on 26 August 2023), and dns-txt (https://github.com/watson/dns-txt, accessed on 26 August 2023). Specifically, "multicast-dns" facilitates the mDNS/DNS-SD related functionalities, "qlobber" is employed for spatial-matching logic, and "dns-txt" is utilized for encoding TXT RR. On the WoT client side, HTTP client operations are handled by node-fetch (https://github.com/node-fetch/node-fetch, accessed on 26 August 2023). For the WoT device, we use "fastify" (https://github.com/fastify/fastify, accessed on 26 August 2023) to manage HTTP interactions efficiently.

In our design, both the capability description and spatial expression are encapsulated in the TXT RR, as formally defined in Equation (11). Here, $\delta_{info}$ refers to capability descriptions, while $\Theta_{SQE}$ and $\vartheta_{SRE}$ represent spatial expressions. When a WoT device receives a TXT RR containing an SQE, it initiates the matching process as delineated in Equation (3). If a match is identified, the device includes an SRE in the response message.

*5.2. Experiments*

Building on the implementation described in the previous section, we carried out four experiments to assess the network traffic of our approach. For these tests, we deployed 10, 20, 30, 40, and 50 light services in individual rounds within a LAN, distributing them evenly across five physical computers, each equipped with an Intel Core i7 processor and 32 MB RAM. While the experiments were not conducted on real embedded computers, we consider this approach to be a valid simulation for our purposes. The focus of these experiments was on network traffic rather than factors like CPU usage or execution time, making the chosen setup a reasonable proxy for assessing our objectives.

Following this, a client was tasked with discovering the services, and we employed a sniffer tool to capture and analyze the messages. Experiment 1 unfolded over a specific period, during which the client multicast query messages to obtain the IP addresses of

matched devices. During this phase, we measured all the message sizes, both for query and response, in bytes. Experiment 2 focused on assessing the total number of messages exchanged within the same timeframe.

The results of Experiment 1 are illustrated in Figure 5. To depict both the successful and failed matching cases, we have categorized the space-aware discovery tasks into three groups based on the match rates: a match rate of 100%, a match rate of 50%, and a match rate of rate 25%. The term "base" refers to the original mDNS/DNS-SD Discovery mechanism that lacks spatial functionalities. A match rate of 100% means that all the devices meet the criteria defined by the spatial expression (SQE), representing an idealized scenario. Conversely, a match rate of 50% implies that only half of the devices meet the criteria, while a match rate of 25% means that a mere quarter of the devices are successfully matched.

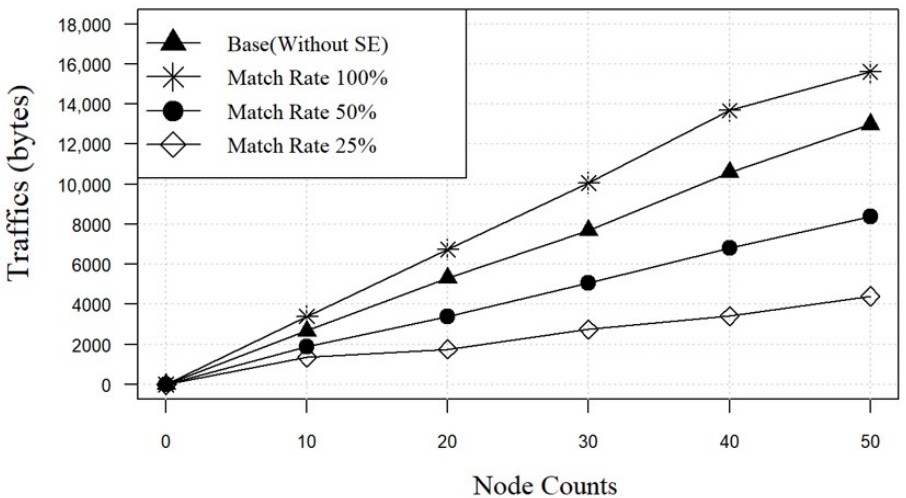

**Figure 5.** The traffics generated when discovering services under for different match rates and number of devices (nodes).

As observed in Figure 5, a match rate of 100% results in the highest traffic, even surpassing the baseline. This increase in traffic is attributable to our approach, which carries additional information, specifically SQE and SRE. Furthermore, a match rate of 100% signifies that all of the services meet the matching criteria, consequently resulting in more response traffic compared to the baseline. As the match rate begins to decrease, the overall traffic correspondingly declines. This reduction occurs as more devices are filtered out and fail to meet the matching criteria, leading to fewer responses and, therefore, less traffic.

The results of Experiment 2 are displayed in Figure 6. The main distinction of Experiment 2 from the previous one lies in the measurement of packet counts rather than the total traffic flow. Interestingly, the number of packets in the baseline is found to be the same as that in the match rate of 100% case. This observation highlights that our approach does not generate additional packets, though it is worth noting that each packet is larger in size.

The objective of Experiment 3 is to compare the system that incorporates spatial information with the baseline version. In this experiment, we inserted 3 to 5 SQEs to observe the traffic flow, measuring the cumulative bytes resulting from query and response messages during the period when the client multicasts query messages to obtain the returned IP addresses of the matched WoT. This experiment was conducted to assess the total overhead incurred when the client wishes to discover multiple spaces simultaneously. We evaluated the overhead by including three to five SQEs in the multicast query message. Theoretically, the LAN's traffic in bytes will increase with the number of embedded SQEs in a single query message. The results, shown in Figure 7, demonstrate that as the number of smart devices increases, the traffic flow follows a descending order: base, SQE = 3, SQE = 4, and SQE = 5. The outcome of the experiment aligns with the theoretical expectations.

Apparently, space-aware discovery introduces additional overhead due to the size or quantity of SQEs.

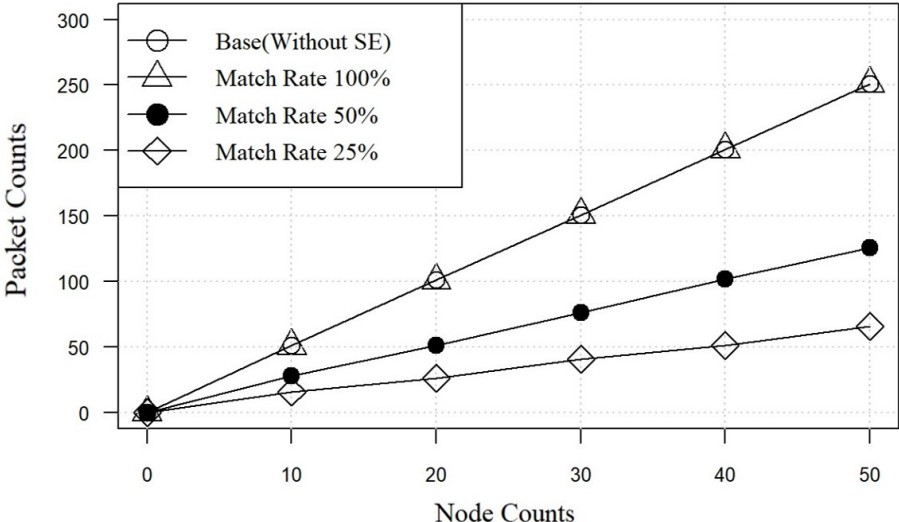

**Figure 6.** The packet counts generated when discovering services under different match rates and numbers of devices (nodes).

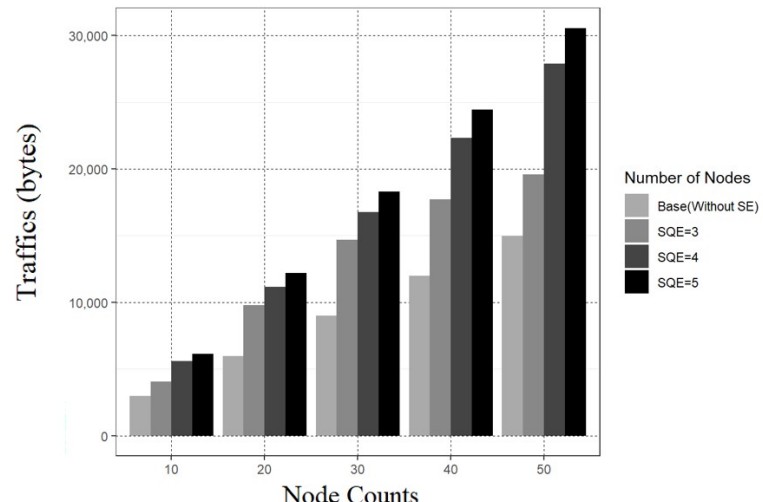

**Figure 7.** The impact of the quantity of SQEs on traffic across various numbers of devices (nodes).

Experiment 4 aims to investigate the effect of device density on latency, exploring various quantities of devices (nodes). In this setup, the x-axis represents the ratio of the number of devices to the number of spaces (i.e., device density), and the y-axis displays the latency in milliseconds, recorded from the time of multicasting query messages to obtaining the IP addresses of all matched devices. We tested four different ratios of 2, 5, 10, and 20 and considered three sets of device numbers: 20, 40, and 60. The results for these are represented by three distinct lines: square, triangle, and circle, respectively (see Figure 8).

Theoretically, latency should increase with the ratio. If a space is divided into more sub-spaces (i.e., the number of spaces on the y-axis increases), there is a higher chance to filter out devices not located appropriately. As a result, fewer devices will be returned, leading to reduced latency and improved performance. Consider an example with 40 lights within one space. In this case, all lights would match a space-aware discovery since their locations align with the search range, yielding a ratio of 40. Conversely, if the same 40 lights are spread across two spaces, $S_1$ and $S_2$, some can be filtered out by adjusting the SQE (e.g., SQE = /root/$S_2$), resulting in only 20 matches. As illustrated in Figure 8, the ratio of

2 yields the lowest time consumption. As the ratio increases (i.e., the number of spaces decreases), the time consumption grows, slowing the response rate. This experiment demonstrates that latency is minimized when the filtering capability of SQE is more refined and when device density is low.

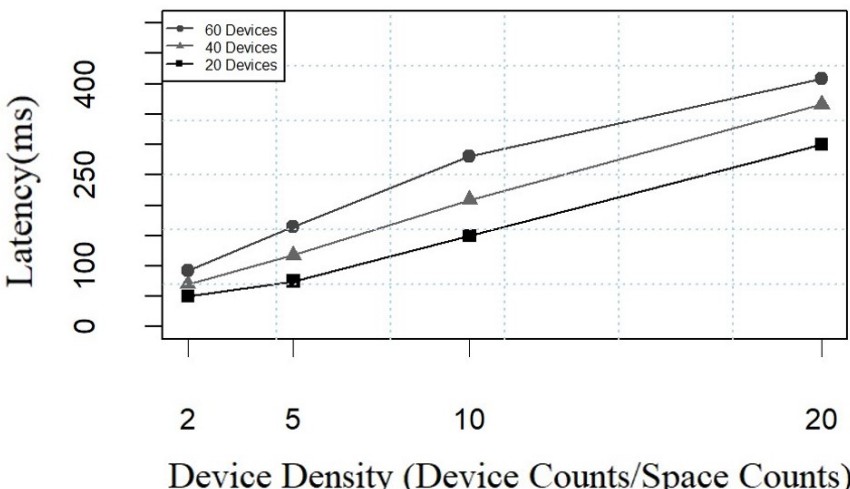

**Figure 8.** The impact of device density on the latency across various numbers of devices (nodes).

*5.3. Discussions*

In this sub-section, we explore the technical considerations, constraints, and limitations, along with the issues that lie outside the scope of our proposed approach.

5.3.1. Complex Spatial Query

The existing SQE framework is designed to handle hierarchical queries, offering support for two kinds of wildcards: "*" for single-level matches and "#" for multiple-level matches. By combining these notations, users can perform a range of advanced search operations. For instance, to find devices near a specific location H, one can employ the multi-level wildcard "#" in an SQE query as follows: "/Home/C/D/#", where D serves as the lowest common ancestor of C and its sibling nodes.

While the current SQE is already quite versatile, it could be further extended with more advanced query operators for more complex searches. However, such advancements would entail a performance trade-off. To elaborate, the concept of "nearby" locations could be incorporated by allowing clients to specify what "nearby" means based on path distances in the ambient structure tree (AST). In this scenario, the lowest common ancestor, denoted by D, could be substituted with a parent node to widen the search scope. For example, the query "/Home/C[@nearBy(2)]" indicates that the search radius extends to two ancestral levels above ambient C. Interestingly, this functionality could be implemented by drawing upon existing languages tailored for tree-like structures, such as XPath [30]. Nevertheless, it is important to recognize that incorporating such features would require additional computational resources.

5.3.2. Matching Efficiency of Spatial Expressions

Our implementation utilizes a Trie data structure, where each node represents a distinct ambient (or space). This structure enables highly efficient searches, with an average time complexity of $O(d)$, where d is the average depth of the AST. However, this efficiency can be compromised when wildcard operators are employed. For instance, a multi-level wildcard would necessitate a traversal of potentially every node in the tree, particularly if a wildcard expression like "#" is utilized.

There are two possible strategies to improve overall performance. One straightforward approach to mitigating computational overhead is by constraining the depth of the AST

and restricting the use of wildcards. By doing so, we can make the search operation more predictable and efficient. Another avenue involves caching mechanisms on both the client and device ends. Such mechanisms could expedite frequent spatial queries. For example, the client could cache the locations of recently searched devices along with a timestamp. Devices, on the other hand, could store recent SQEs that include multi-level wildcards, along with their corresponding outcomes. While caching introduces additional storage requirements, its primary trade-off would be the potential overhead associated with cache misses.

### 5.3.3. System Initialization

When initializing the system, it is essential to configure the AST according to the specific floor plan of the environment. In practice, the hierarchy of spaces can be tailored to the user's needs, allowing for a flexible division of areas. For instance, the living room might be subdivided into multiple sub-spaces, reflecting the likelihood of having more devices and services located there. Additionally, the implementation of space-aware service discovery model requires the assignment of an SRE for each service. This step can be a burden for device/service developers.

### 5.3.4. Location Update and Dissemination

Although the real-time automatic detection and updating of device locations are intriguing aspects, they fall outside the scope of this paper. This work assumes that home appliances and sensors hosting nodes remain largely stationary once deployed. Should a device be relocated from one space to another, the device must be designed to recognize its new location and adjust the SRE accordingly and automatically. While solutions like BLE-based positioning are viable, they come with trade-offs—specifically, the need for each device to be equipped with a BLE chip. Alternatively, IoT deployment tools could be developed to update and disseminate location information to the devices, thereby alleviating the burden on users.

### 5.3.5. Working with LLNs

The current approach is designed based on mDNS/DNS-SD, operating on top of the IP layer. Consequently, the scope of the approach is restricted to managing spatial information within an IP network. However, this method can be extended to LLNs, such as IEEE 802.15.4 [31], commonly used by IoT devices. This can be achieved through integration with a 6LoWPAN gateway, which constructs an UDP/IP layer on top of IEEE 802.15.4. BLE is another prevalent LLN for IoT devices. We have successfully devised strategies for seamlessly managing devices in both 6LoWPAN and BLE using mDNS/DNS-SD. These achievements make the approach proposed in this paper applicable to two of the most popular LLN standards utilized by IoT devices.

### 5.3.6. Interoperability

To facilitate the spatial awareness of legacy devices, their software or firmware needs to be updated to process spatial expressions. However, we recognize that updating all legacy devices to be space-aware may not be feasible in real-world scenarios. To address this, our approach is designed to be backward-compatible with non-spatial-aware clients and devices by using TXT RR in DNS. Originally intended for human-readable text, TXT RRs are now also commonly used to store machine-readable information. Importantly, information stored in a TXT RR is ignored by receiving endpoints that are unfamiliar with its format. In this manner, when a legacy client that is not spatially aware receives a TXT RR containing a spatial expression, it simply disregards the message. Likewise, an IoT device that is not space-aware will ignore the TXT RR if it encounters an SQE and fails to recognize its format. Therefore, our approach is compatible with existing mDNS/DNS-SD implementations, requiring no modifications to function alongside non-spatial-aware clients and devices.

Regarding service discovery, our approach is also compatible with devices that adhere to the CoRE (constrained RESTful environments) standard. When IoT devices offer services via HTTP or CoAP (constrained application protocol), the service descriptions, encoded in CoRE Link Format, can be transmitted using TXT RR. While CoRE does not define a standard attribute for location, a practical approach would be to introduce a custom link extension, denoted as "se=", where the SQE can be placed following the equal sign.

## 6. Conclusions

In this study, we introduced a non-intrusive extension to mDNS/DNS-SD, enabling the space-aware service discovery model. This mechanism has been formally specified to facilitate both performance analysis and implementation. A series of experiments were conducted to verify the performance of the proposed mechanisms, showing promising results. Current experiments are conducted on PC-based hosts since the experiments are primarily I/O-bound rather than CPU-bound. However, considering the importance of real-world testing on resource-constrained nodes, we will evaluate our approach on actual embedded/IoT devices.

Our method still has areas for further refinement. Currently, the system assumes that the location of each WoT device is predefined and remains fixed, a limitation we aim to overcome by automatically updating the SRE through BLE positioning in future iterations. Additionally, we recognize that our current implementation does not address security concerns. Given that the information within the TXT key = value pairs may be vulnerable to interception, we plan to explore robust and efficient methods such as DNSSEC (DNS security extensions) [32] for encrypting the RRs. This continued work will enhance the safety and completeness of our approach, moving it closer to real-world applicability. Lastly, we plan to develop a set of user-friendly tools equipped with intuitive graphical user interfaces to simplify the process of setting up and configuring user preferences.

**Author Contributions:** Conceptualization, C.-F.L.; methodology, C.-F.L.; software, Y.-J.W.; validation, Y.-J.W. and C.-F.L.; formal analysis, C.-F.L.; data curation, Y.-J.W.; writing—original draft preparation, Y.-J.W. and C.-F.L.; writing—review and editing, C.-F.L.; visualization, Y.-J.W.; supervision, C.-F.L.; project administration, C.-F.L.; and funding acquisition, C.-F.L. All authors have read and agreed to the published version of the manuscript.

**Funding:** This work is partially sponsored by National Science and Technology Council, Taiwan, under grant 112-2221-E-004-001.

**Data Availability Statement:** Not applicable.

**Conflicts of Interest:** The authors declare no conflict of interest. The founding sponsors had no role in the design of the study; in the collection, analyses, or interpretation of data; in the writing of the manuscript; or in the decision to publish the results.

## Abbreviations

The following abbreviations are used in this manuscript:

| | |
|---|---|
| AST | Ambient structure tree |
| GPIO | General purpose input/output |
| LAN | Local area network |
| LLN | Low-power and lossy network |
| mDNS/DNS-SD | Multicast domain name system/DNS-based service discovery |
| QoS | Quality of service |
| RR | Resource record |
| SE | Spatial expression |
| SQE | Spatial query expression |
| SRE | Spatial response expression |
| WoT | Web of things |

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
