# Peer review of "Enabling Space-Aware Service Discovery Model in Home Networks through a Compatible Extension to mDNS/DNS-SD"

_electronics, doi:10.3390/electronics12183885_

Round 1

Reviewer 1 Report

Technical Questions:

  1. How are the spatial expressions generated initially? Is this manual configuration or automated?
  2. Could the proposed techniques be extended to mobile devices whose locations change dynamically?
  3. What are the limitations in terms of scalability or network overhead when using more complex spatial queries?
  4. How is matching efficiency and latency impacted when there are a very large number of subdivided spaces?
  5. Have authors considered supporting more complex spatial query expressions, such as querying for spaces that are adjacent/nearby rather than strictly hierarchical? This could allow more flexible discovery.
  6. How do authors handle devices that may move to new locations, rather than remaining static? Does the system allow for location updates?
  7. Could the spatial expressions be encrypted or obfuscated in some way to improve privacy/security?
  8. Does supporting the spatial expressions require changes to existing mDNS/DNS-SD implementations or can it work with unmodified versions?

Suggestions:

  • Provide more implementation details on the node.js prototype - what modules were used, how messages were passed, etc. This could help readers reproduce it.
  • Consider evaluating the approach on actual embedded/IoT devices rather than just PCs to better validate performance on resource-constrained nodes.
  • Investigate how the techniques could be expanded to support discovery in low-power wireless protocols like BLE, ZigBee, etc. beyond just IP-based networks.
  • Analyze the computational overhead on devices for matching spatial expressions compared to standard mDNS/DNS-SD.
  • Discuss if any indexing or caching techniques could help speed up frequent spatial queries.
  • Compare your spatial expressions to other semantic location models like CoRE Link Format to see if they could be aligned.

Reviewer 2 Report

The proposal presents a seamless approach to integrate space and location information with mDNS/DNS-SD, a widely utilized service discovery protocol in home networks. The authors present a formal specification for this approach, develop a prototype, and perform a series of experiments to evaluate the efficacy and potential of the proposed method. The proposal presents an interesting topic; however, the following aspects were identified: 

1.     In the last paragraph of the introduction, in addition to indicating the objectives of the proposal, it is suggested to indicate the main scientific contribution. Also, it is suggested to add a paragraph where the structure of the proposal is indicated and described.

2.     It is suggested to update the review of the state of the art since all the related works presented are NOT from the last five years. In addition, after presenting the related works, not only the similarities but mainly the differences should be indicated in order to highlight the novelty of the proposal.

3.     It is suggested that when acronyms are used the first time they appear in the text, their meaning should be specified and then only the corresponding acronym should be used. Therefore, it is suggested to specify the meaning of the acronyms QoS, PTR, SRV, TXT, A, and AAAA. In addition, it is suggested to check that this is not the case with the other acronyms used.

4.     It is suggested to improve the quality of figure 4 for a better reading of the texts and compression of the figure.

5.     Is it intended that the results obtained in this proposal allow in the future its application in the real world, i.e., the resident of the house could identify on a screen or mobile device the location of the IoT devices according to their preferences and comfort? If the answer is yes, it is suggested to indicate it in the future work.

6.     If the answer to the previous comment is yes, it is suggested to include in the review of the state of the art some works related to the IoT on the discovery of the behavior, preferences or tastes of the residents in a smart home; as well as some work on the monitoring and control of smart devices.

Not applicable

Round 2

Reviewer 1 Report

Authors have adequately addressed all of the points I highlighted in my review. Specifically:

  • The related work section has been thoroughly revised to include more recent and relevant studies, with differences from their work clearly outlined.
  • The introduction and conclusion now clearly state the key objectives and highlight the main scientific contributions of their approach.
  • Acronyms are properly defined upon first use, and a list of abbreviations has been added.
  • The overall clarity and quality of the diagrams and figures has been improved.

Author Response

The authors extend their sincere gratitude to the reviewer for offering insightful comments that have significantly enhanced the quality of this manuscript.

Reviewer 2 Report

It is noted that the authors have updated section 2 with more recent related work, however, it is necessary to clearly indicate the main differences of the proposal with respect to all the related work presented in order to understand the novelty of the proposal.

NA

Author Response

Modification

A new paragraph has been added at the end of Section 2 to clearly delineate the key differences and advancements in comparison to existing literature.

Response

We would like to express our gratitude to the reviewer for highlighting this issue. The primary distinctions can be categorized into three key aspects.

First, while the research trends largely aim to optimize efficiency and reduce network traffic in mDNS/DNS-SD implementations, extend these protocols for LLN, or focus on domain-specific applications, our work takes a divergent path. Specifically, our research focuses on enhancing the spatial search functionalities within mDNS/DNS-SD. To this end, we propose an innovative extension that seeks to elevate the quality of service discovery in home networks through the integration of spatial data.

Second, as mentioned above, Stolikj et al. have proposed two separate strategies for augmenting QoS attributes. We employ both of these strategies to craft a more nuanced and effective service description. To elaborate, we augment the domain name with service types denoted as Sub-Types, aligning with their first recommendation. Concurrently, we incorporate spatial information within the TXT resource records, utilizing a Key=Value schema in accordance with their second recommendation. This dual-strategy approach facilitates a more comprehensive service discovery mechanism.

Lastly, when it comes to the formal abstractions of ambient environments, our research distinguishes itself from prior studies  that primarily focus on modeling tangible objects or well-defined spaces. In contrast, our work places a premium on capturing the spatial relationships between different locations, integrating this aspect seamlessly into the existing service discovery protocols (namely, mDNS/DNS-SD). This allows for a more holistic and applicable approach to service discovery, tailored to the complex spatial dynamics of modern home networks.